

# 1 Cloud condensation Nuclei over the Southern Ocean: wind
# 2 dependence and seasonal cycles

John L. Gras[1] and Melita Keywood[1]
[1]Oceans and Atmosphere, CSIRO, Aspendale, 3195, Australia
*Correspondence to*: Melita Keywood (melita.keywood@csiro.au)
**Abstract.** Multi-decadal observations of aerosol microphysical properties from regionally representative sites can be used to
challenge regional or global numerical models that simulate atmospheric aerosol. Presented here is an analysis of multi-decadal
observations at Cape Grim (Australia) that characterise production and removal of the background marine aerosol in Southern
Ocean marine boundary layer (MBL) on both short-term weather-related and underlying seasonal scales.
A trimodal aerosol distribution comprises Aitken nuclei (< 100 nm), CCN/accumulation (100-350 nm) and coarse mode
particle (> 350 nm) modes, with the Aitken mode dominating number concentration. While the integrated particle number in
the MBL over the clean Southern Ocean is only weakly dependent on wind speed the different modes in the aerosol size
distribution vary in their relationship with windspeed. The balance between a positive wind dependence in the coarse mode
and negative dependence in the accumulation/CCN mode leads to a relatively flat wind dependence in summer and moderately
strong positive wind dependence in winter. The change-over in wind dependence of these two modes occurs in a very small
size range at the mode intersection, indicative of differences in the balance of production and removal in the coarse and
accumulation/CCN modes.
While a marine biological source of reduced sulfur appears to dominate CCN concentration over the summer months
(December to February) other components contribute to CCN over the full annual cycle. Wind-generated coarse mode sea-
salt is an important CCN component year round and is the second most important contributor to CCN from autumn through to
mid-spring (March to November). A portion of the non-seasonal dependent contributor to CCN can clearly be attributed to
wind generated sea-salt with the remaining part potentially being attributed to long range transported material. Under
conditions of greater supersaturation, as expected in more convective cyclonic systems and their associated fronts, Aitken
mode particles become increasingly important as CCN.

## 25 1 Introduction

There can be little doubt that global numerical models that include realistic aerosol microphysical processes, coupled with
accurate data on aerosol and precursors are the best tools to understand future aerosol impacts on climate, as attested by
developments such as Spracklen et al. (2005), Pierce et al. (2006), Korhonen et al. (2008), Wang et al. 2009, Mann et al.
(2010) and Lee et al. (2015). While one significant component in the building of confidence in such models is the availability



of representative climatologies of aerosol properties, such as cloud condensation nucleus (CCN) concentration on a global
scale, information on the range of aerosol properties on this scale, certainly from in-situ observations, is always likely to be
relatively limited.   Multi-decadal observations of a wider range of properties from regionally-representative sites provide a
complementary basis for challenging these models, including for example dependencies on meteorology, seasonal and inter-
annual variations.   With continuing refinement of microphysical representations within climate models there are opportunities
to examine these more subtle features of the aerosol within its very dynamic relationship in the weather-climate system, as
well as refine the understanding of the various sources that contribute to aerosol regionally.
Work reported here draws on multi-decadal observations at Cape Grim (Australia) to characterise aspects of the clean marine
aerosol related to production and removal in the Southern Ocean marine boundary layer (MBL) on two different timescales.
The first is short-term weather-related and the second is a re-examination of underlying seasonal-scale variations.   Particles
examined include two broad populations, Aitken nuclei and CCN.  Aitken nuclei are represented by N3 and N11, where N3 is
the concentration integrated across all particle diameters greater than 3 nm and N11 likewise for particle diameters greater
than 11 nm (for these observations there is a practical upper size limit of 10 μm). This population is usually dominated by
particles in the Aitken mode, which in this MBL has a number distribution mode diameter around 20 nm.   CCN is the
population of particles active at supersaturation levels typical of clouds; for this work the population examined mainly
comprises particles active at a supersaturation of 0.5% (CCN0.5) and at 0.23% (CCN0.23); the calculated lower particle size
threshold is composition dependent but typically for CCN0.5 includes particles larger than approximately 50 nm diameter and
for 0.23% around 78 nm.  These particles comprise a significant fraction of the MBL accumulation or cloud-processed mode.
One very characteristic feature of aerosol in the Southern Ocean MBL is a strong and persistent underlying seasonal cycle that
in many aspects resembles a seasonal "pulse" over the summer months (December, January and February).  This underlying
pattern of seasonal variation was used for example by Bigg  et al. (1984), as evidence linking MBL N3 particle population to
solar radiation and a probable free troposphere source.  Seasonal covariance of aerosol and methane sulfonic acid (MSA), an
aerosol phase oxidation product of dimethyl sulphide (DMS) was also used by Ayers  and Gras (1991) and Ayers  et al. (1991)
as evidence supporting a major role for marine biogenic reduced sulfur source in driving the CCN seasonal concentration
cycle.  This CCN subset of the aerosol population has the ability to modify cloud physical and optical properties across a
substantial fraction of the relatively pristine Southern Ocean area, a potential climate impact first identified by Twomey (1974).
The likely significant role of marine biogenic sources providing precursors for secondary aerosol in the remote MBL has led
to suggestion of potential feedback mechanisms, particularly the CLAW hypothesis by Charlson  et al. (1987).  CLAW, named
after the author's initials, proposes regulation of global temperature by DMS emission, through CCN-driven cloud albedo
feedback. Despite considerable development in understanding chemical processes and transport, as well as recent studies
extending earlier surface-based seasonal coherence studies to wider areas through satellite remote sensing (e.g. Gabric  et al.,
2002, Vallina  et al., 2006), the relatively limited progress in validation of the CLAW hypothesis as summarised, for example





by Ayers and Cainey (2007) and referred papers, illustrates the extreme complexity of processes controlling particle number
population in the remote MBL. Quinn and Bates (2011) go further, arguing that the relatively simplistic feedback mechanism
via DMS emission as represented in the CLAW hypothesis probably does not exist.
The overall response of natural production mechanisms in this region to climate change and even increasing anthropogenic
sources is still relatively uncertain and contributes to the overall uncertainty in future climate prediction due to indirect aerosol
forcing as illustrated in successive IPCC reports, recently by Boucher et al. (2013). The purpose of the present work is to
examine, or re-examine some characteristic features in the MBL aerosol at the relatively pristine site of Cape Grim, that should
provide a useful challenge and resource for the continually developing numerical models addressing these uncertainties.
**2 Methods -site description and instruments**
The Cape Grim baseline atmospheric program is the principal Australian contribution to the WMO Global Atmosphere Watch,
with an observatory located at the northwest tip of Tasmania (40° 41' S, 144° 41' E). Situated on a cliff and 94 m above sea
level the location maximises observation of Southern Ocean air that has had minimal recent anthropogenic impact. Air sampled
in the "Baseline" sector (190° — 280°) typically traverses several thousand kilometres of Southern Ocean since previous land
contact. The location of Cape Grim and the Baseline wind sector are shown in Fig. 1.
Measurement of airborne particles at Cape Grim commenced in the mid 1970s and generally sampling has followed what are
now WMO GAW Aerosol program recommendations (WMO 2003). N3 is effectively the total particle concentration and
concentrations reported here were determined using a TSI Ultrafine Condensation Particle Counter (CPC). N11 was
determined using TSI 3760 and TSI 3010 CPCs, CCN concentration for particles active at various supersaturations, but
primarily at 0.5% supersaturation, were determined using an automated static thermal gradient cloud chamber (Gras 1995).
Calibration of the CCN counter utilised a CPC (TSI 3760 / TSI 3010) and monodisperse particles of ammonium sulfate or
sodium chloride. The Koehler equation and van't Hoff factor approach were used to compute solute activity, as reported by
Low (1969). Size distribution data were obtained using an ASASP-X laser single particle size spectrometer, calibrated using
polystyrene latex spheres and corrected for refractive index to m=1.473-0i. A TSI 3790 mobility analyser with CPC (TSI 3010)
was also used to measure size distribution. Volatility measurements were made using a high temperature quartz tube furnace
with the CCN and CPC counters. Other reported measurements include Cape Grim methane sulfonic acid (MSA) and sea-
salt, from PM10 high-volume filter samples dimethyl sulfide (DMS), as described by Ayers and Gillett (2000) and estimates
of surface UV radiation based on satellite and ground based observations. UV radiation data, as midday erythemal irradiance
(250-400 nm, weighted) were obtained from the NASA TOMS satellite record for the region upwind of Cape Grim (40-45 ºS
110-130 ºE) available from http://macuv.gsfc.nasa.gov/. UV irradiance values were also computed using ozone column data
for Melbourne (38 °S 145 °E), determined by the Australian Bureau of Meteorology using the empirical model of Allaart et


al. (2004).  For the available overlap of the two UV irradiance records the correlation coefficient of monthly values is r2=0.99
(n=290).

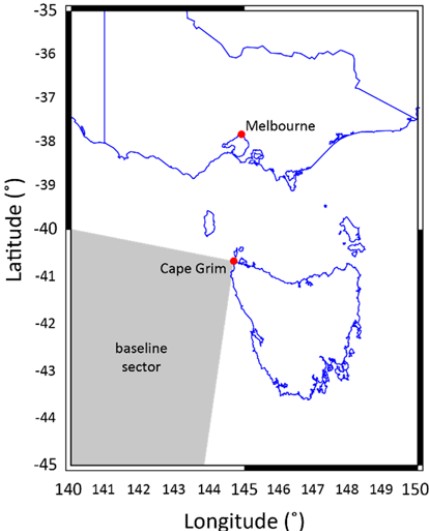


**Fig. 1. Map of south-eastern Australia and Tasmania, showing the location of the Cape Grim Baseline Air Pollution Station and**
**clean air "Baseline" sector.**
**3 Short term processes and CCN concentration**
The Southern Ocean region upwind of Cape Grim comprises part of the "roaring forties", and has a well-earned reputation for
strong and persistent winds.  For 1999-2006, for example, median wind speed for the Baseline maritime sector was 10.7 ms$^{-1}$,
the 5$^{th}$ percentile was 3.4 ms$^{-1}$, and the 95$^{th}$ percentile 19.1 ms$^{-1}$.  Most primary particle sources over the Southern Ocean can
be expected to be wind-dependent such as spray and bubble processes, and at higher wind speeds windshear or spindrift; in
addition wind is important for dispersal.

The background particle size distribution in the MBL at Cape Grim is typically tri-modal and the two modes of most obvious
interest from a CCN perspective for summer are shown in Fig. 2a. Baseline conditions (local wind direction 190-280º) were
selected using ambient radon concentration of less than 150 mBeq m$^{-3}$; this includes approximately 80% of Baseline sector
observations.  Concentration data were derived using an ASASP-X size spectrometer for December, January February 1990-
1995 and are plotted as median concentrations over 6 ms$^{-1}$ wind bands.





Cumulative concentrations for particles with diameters between ~117 nm - 350 nm, D > 350 nm and D > 1000 nm for the
same data are shown in Fig. 2b.

Concentration in the coarse mode size range increases with wind speed (Figs. 2a and 2b); this is a widely reported phenomena
resulting from bubble and spray mechanisms of wind generation of sea-salt particles for which there are a number of numerical
parameterisations, see for example Gong  (2003).  Wind generated sea salt is normally the most abundant aerosol mass
component over the Southern Ocean,  with typical PM10 sea-salt mass loadings at Cape Grim around  12 $\mu g\ m^{-3}$.  This coarse
salt mode dominates the volume distribution throughout the year, although in number concentration the mode contribution is
small; as shown, for example by Covert et al. (1998) the contribution to CCN0.5 in late spring 1995 was around 16%.  The
particle number concentration for the coarse mode was determined by fitting log-normal volume distribution functions for
ASASP-X data in selected (6 $ms^{-1}$) wind bands for data collected during 1991-1994. For winter, combination of these observed
representative size distributions with the more complete record of wind dependent concentrations for D > 350 nm yields an
overall median number concentration of 26.6 $cm^{-3}$ for the coarse mode.  During the summer the coarse mode provides only a
small fraction of particles in the CCN size range.  Early work by Gras and Ayers (1983) using electron microscopy of individual
particles, found that typically 1% of particles at 100 nm were clearly identifiable as sea-salt at Cape Grim in summer. While
considerable work has been reported on understanding wind-dependent generation of sea salt, some aspects including the
distribution of sea salt for sizes less than around 100 nm are still quite poorly characterised (Bigg  2007).

In contrast to the increase in particle concentration in the sea-salt dominated coarse mode, in summer the 115-350 nm size
range particle concentration decreases with increasing wind speed (Fig. 2b). This size range in the Southern Ocean MBL is
part of a size mode centred at about 120 nm that is strongly associated with CCN. Although this mode is generally referred to
as the accumulation mode, given its importance in cloud processing in the MBL it is identified here as the accumulation/CCN
mode.  Particles in this mode were shown to be volatile by Hoppel et al. (1986), who proposed an in-cloud growth mechanism
involving aqueous-phase photochemical oxidation of gaseous precursors together with the physical processes of diffusion and
coalescence.
As shown in Figs. 2a and 3 the accumulation/CCN mode is strongest in summer and although it is also present in winter, it is
often vestigial and difficult to separate from the coarse mode in single particle distributions.
The evidence of negative wind speed dependence in the accumulation/CCN mode amplitude for current-hour wind-speed in
summer is an interesting feature of this mode, as is the very sharp changeover from a positive relationship with wind speed in
the coarse mode to the negative relationship in the accumulation/CCN mode, occurring at around 350 nm diameter at the mode
intersection and within a narrow range of less than 50 nm.



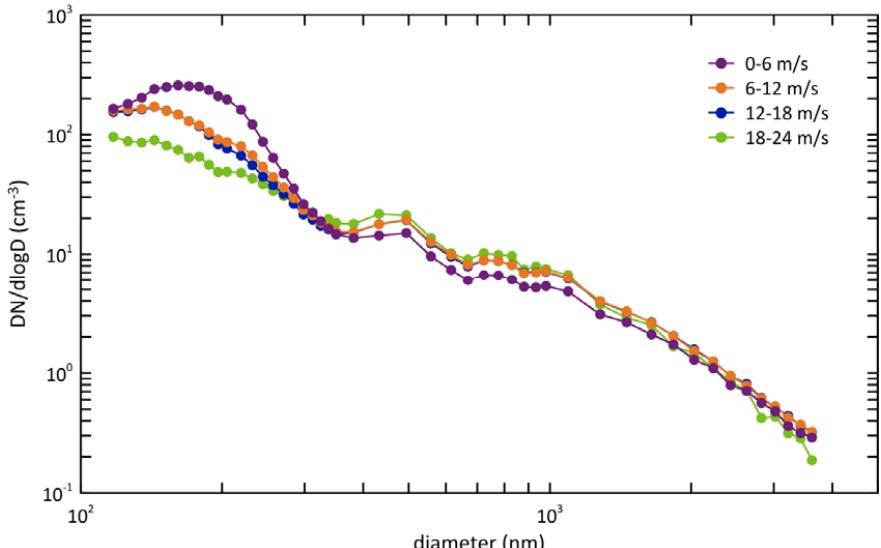

**Fig. 2a. Summer clean MBL size distributions in 6 ms⁻¹ wind bands, medians of hourly data**

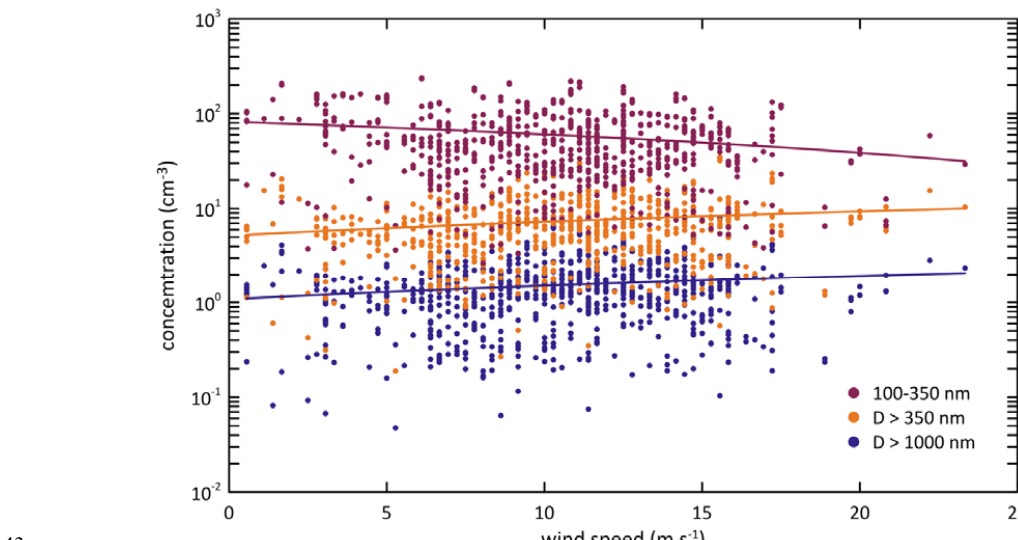

**Fig. 2b. Cumulative concentration as a function of wind speed for accumulation/CCN mode (117 nm –350 nm) and for D > 350 nm**
**and D > 1000 nm.**





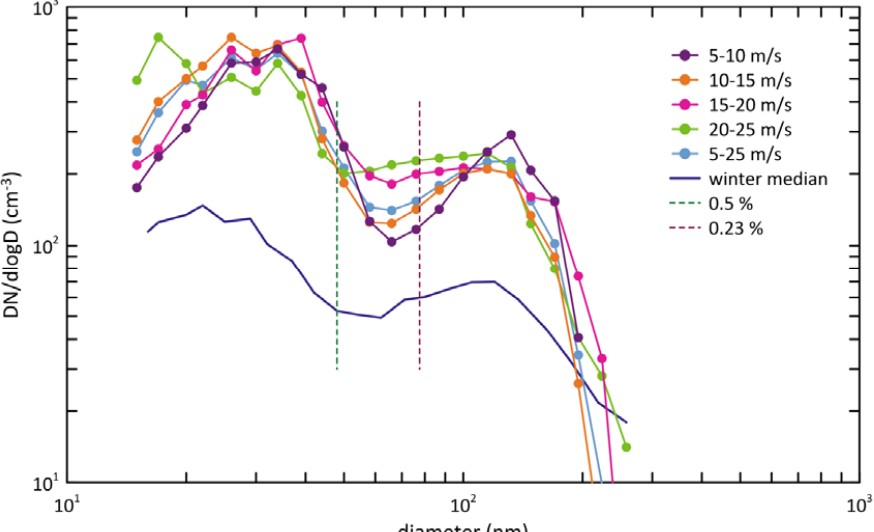

**Fig. 3. DMA derived size distribution including Aitken-accumulation/CCN mode gap as a function of wind speed.**
**3.1 Wind strength and CCN concentration**
A more extensive series of concentrations of CCN, N3 and N11, here covering the "Baseline" sector conditions for 1999-2006
gives more information on wind speed dependence in these integral measures, and allows some inferences on mode behaviour.
For summer, with selection of Baseline conditions using radon concentrations < 150 mBeq m$^{-3}$, the CCN0.5 dependence on
wind speed overall is negative and represents a change for the median of about -14 cm$^{-3}$ at 20 ms$^{-1}$ (from 121 cm$^{-3}$ at zero wind
speed), while additional restriction to wind speed greater than 7 ms$^{-1}$ changes this dependence to positive overall and represents
+9 cm$^{-3}$ at 20 ms$^{-1}$; this is shown in Fig. 4.
Subtracting the coarse mode wind dependence determined from the ASASP size spectrometer results in much stronger negative
wind dependence for CCN0.5. This is predominantly due to the accumulation/CCN mode and now represents a change of
close to -60 cm$^{-3}$ at 20 ms$^{-1}$ for the median. Restriction to wind speeds greater than 7 ms$^{-1}$ reduces this dependence slightly,
giving a change of around -51 cm$^{-3}$ at 20 ms$^{-1}$.
Particle concentrations increase rapidly with decreasing wind speed for speeds less than around 7 ms$^{-1}$ and when more relaxed
radon baseline criteria are applied, indicating the influence of local coastal effects and particularly recirculation of air that has
had recent land contact. Only minor improvement in rejection of potentially land-contaminated samples results from radon
selection threshold less than 150 mBeq m$^{-3}$, although added screening to exclude low wind speeds is preferable.



In winter for wind speed greater than 7 ms$^{-1}$, CCN0.5 concentration increases with increasing wind speed. Fig. 5 shows the
population of CCN selected for radon concentration less than 150 mBeq m$^{-3}$ and wind speed > 7 ms$^{-1}$, and the change for 20
ms$^{-1}$ for the median's trend is around 25 cm$^{-3}$ (from a reference level of ~31cm$^{-3}$ at zero wind speed). As in the case of summer,
subtraction of the wind-dependence for the coarse mode results in a moderately strong negative trend for the
accumulation/CCN mode component, representing a change of -12 cm$^{-3}$ at 20 ms$^{-1}$.
Figure 5 includes the wind-dependent concentration of CCN active at 0.6% supersaturation observed by Bigg et al. (1995)
from ship-based measurements over the far Southern Ocean (50-54 °S) around mid-winter, under conditions where
photochemical production should be minimal. The wind dependence of the concentration of coarse mode particles at Cape
Grim, utilising fitted wind-dependent volume size distributions from data collected with an ASASP-X single particle size
spectrometer during winters over 1991-1994 and as described earlier, is also given in Fig. 5. In addition, non-volatile aerosol
concentrations determined in three 5 ms$^{-1}$ wind bands, at Cape Grim over two weeks in different winters are shown. These
volatility measurements showed no systematic difference between 350 °C and 900 °C, providing some evidence that the
relationship observed by Bigg et al. (1995), the Cape Grim coarse mode and a significant proportion of CCN at 0.5%
supersaturation at Cape Grim in winter, for high wind speeds, are composed of primary sea-salt.
The expected principal removal mechanism in the CCN size range is nucleation scavenging, (Gras 2009), and dominance of
this loss mechanism on a regional scale is supported by the statistical analysis by Vallina et al. (2006). Nucleation scavenging
should be equally effective across both accumulation/CCN and coarse modes although removal in the coarse mode is masked
by wind-related primary production; in addition some offset in the accumulation/CCN mode is likely from increased transfer
of DMS at the ocean surface. Other microphysical loss processes include washout, "dry" deposition to the surface through
increased spray generation with subsequent scavenging, and increased diffusion to large aerosol surface area, although none
of these removal mechanisms readily explains the rapid reversal of wind dependence around 350 nm diameter. The negative
wind dependence in the accumulation/CCN mode and reversal in dependence around 350 nm are consistent with competing
processes to DMS oxidation modulating the strength of the accumulation/CCN mode. This includes heterogeneous reaction
of SO$_2$ on sea-salt spray as suggested by Sievering et al. (1991), although the chemical mechanism leading to enhanced sulfate
production on spray aerosol has been debated (Laskin et al. 2003, Keene et al. 2004, Sander et al. 2004, von Glasow 2006).
Another possible mechanism is a sink for H$_2$SO$_4$ on super-micron salt aerosol (Yoon and Brimblecombe 2002). Both these
mechanisms could potentially reduce the availability of reactive gases for in-cloud processing, and contribute to negative wind
dependence of accumulation/CCN mode particle concentration.
Over the Southern Ocean, increased wind speed is usually associated with more intense cyclonic systems, accompanied by
increased convection, cloudiness and precipitation probably providing the major link between accumulation/CCN mode and
wind speed. On a local scale at Cape Grim, monthly wind speed and rainfall data are only weakly positively correlated ($r^2$ =
0.12, Baseline) and the correlation for hourly data e.g. for summer using past 12h accumulated rainfall and current wind speed
(excluding within 12 h of a front) is even weaker ($r^2$ = 0.0006). This is a weakness of fixed site observations which reveal only
the immediate state of processes along a Lagrangian pathway taken by an air parcel.





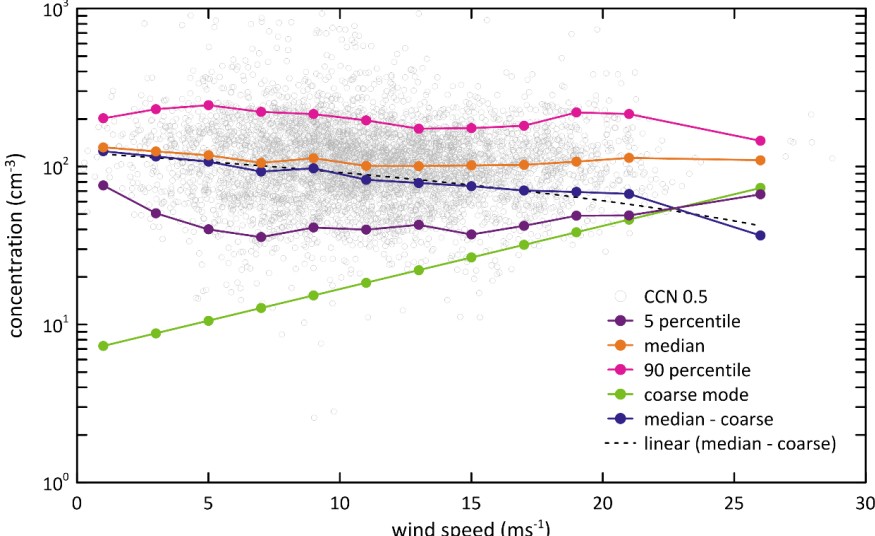

**Fig. 4. CCN0.5 concentration as a function of wind speed, summer, radon < 150 mBeq m⁻³.**

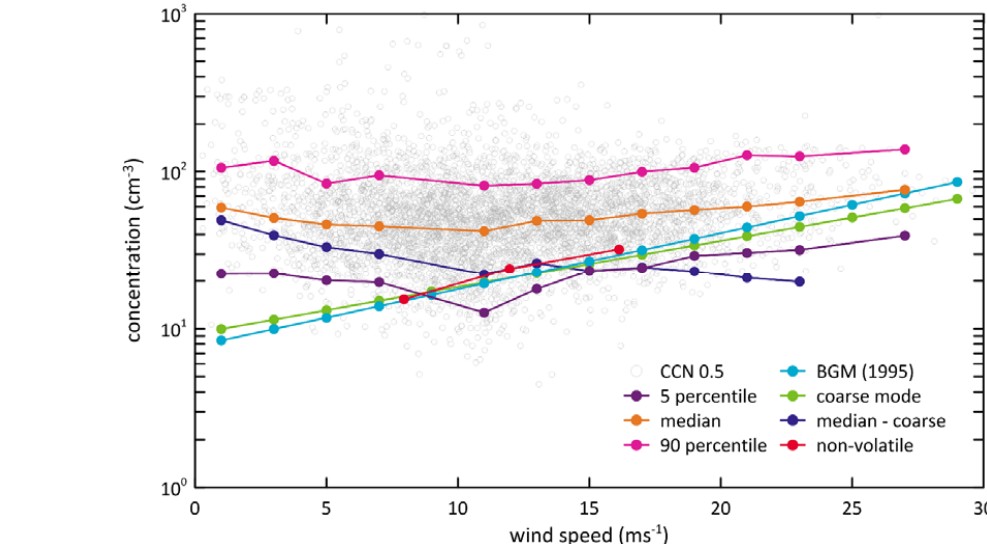

**Fig. 5. CCN0.5 concentration as a function of wind speed, winter, radon < 150 mBeq m⁻³. BGM (1995) is the CCN active at 0.6%**
**supersaturation observed by Bigg et al. (1995).**



Mobility-derived particle size distributions for Cape Grim summers show some evidence of increased convection associated
with stronger winds (Fig 3). As suggested by Hoppel et al. (1986), the dip between the Aitken and cloud-processed modes are
a measure of the smallest particles that nucleate cloud droplets for this environment. For the cases considered (i.e. 2126
summer distributions), the minimum between the modes decreases from around 66 nm at 5-10 ms$^{-1}$ to around 50 nm at 20-25
ms$^{-1}$ and the spread in particle sizes broadens to smaller sizes such that over all conditions the mode diameter is around 115
nm while for 20-25 ms$^{-1}$ concentrations are relatively flat with a broad mode between 60 nm and 130 nm. These values also
illustrate the typical supersaturation levels for this environment; assuming an ammonium bisulphate/MSA mixture, the typical
supersaturation experienced (calculated using Kohler theory) would be somewhat greater than 0.15%, since mode size
represents typical particles exiting clouds. The mode of particles passing through the cloud process extends down to 50 – 66
nm (0.45% - 0.3% supersaturation) depending on environmental conditions (indicated by wind strength). Under the more
convective conditions a much greater fraction of CCN active at 0.5% would be classified as being in the upper end of the
Aitken mode. The instrumental value of 0.5% supersaturation used as the primary measure of CCN at Cape Grim provides a
conservative lower size bound that captures the accumulation/CCN mode under most environmental conditions, albeit at the
expense of including a small and variable fraction of Aitken mode particles.
Particle concentration integrated across the full size spectrum does not show strong wind dependence, for example the
relationship between N11 and wind speed in summer (for data selected with ws > 5 ms$^{-1}$ and radon < 150 mBeq m$^{-3}$) is weakly
positive, representing an overall increase of around 54 cm$^{-3}$ at 20 ms$^{-1}$ or 11% (from a concentration of 482 cm$^{-3}$ not related to
wind speed). In winter with the same selection criteria the trend is around +56% against a reference of 104 cm$^{-3}$, although this
represents a similar concentration increase, in this case 58 cm$^{-3}$ at 20 ms$^{-1}$. These trends are close to those expected from the
coarse mode and subtraction of the parameterised coarse mode leaves the overall trend across the Aitken and accumulation
modes relatively small, at -6 cm$^{-3}$ in summer and +20 cm$^{-3}$ in winter (for a wind strength change of 20 ms$^{-1}$). Using the
difference in N11 and CCN0.5 concentrations and the same wind speed and radon selection criteria as a measure of the
(integrated) Aitken mode concentration indicates a positive dependence in summer of +40 cm$^{-3}$ at 20 ms$^{-1}$, relative to 364 cm$^{-3}$
not related to wind speed change, and in winter a similar concentration increase but greater fractional change of +26 cm$^{-3}$ at
20 ms$^{-1}$ (relative to 67 cm$^{-3}$ not related to wind speed change) for the Aitken mode.
This shift in balance between production and removal for the Aitken mode with increasing wind speed could potentially include
wind-generated sub-50 nm diameter primary sea-salt particles (e.g. Clarke et al., 2006), although it is also entirely consistent
with enhanced free troposphere-MBL exchange from a dominant free-tropospheric particle number source, following the
mechanism summarised for example by Raes et al. (2000).
The concentration of nanoparticles defined here as N3-N11 also increases with wind speed, for data with the same wind speed
and radon selection criteria described above. For summer the concentration increase for a 20 ms$^{-1}$ wind speed increase is
around +8 cm$^{-3}$ and in winter 12 cm$^{-3}$.
While overall the integrated particle number in the MBL over the clean Southern Ocean is only weakly dependent on wind
speed, the wind dependence of the different modes that comprise the full aerosol spectrum is complex. For CCN the balance



between a positive wind dependence in the coarse mode and negative dependence in the accumulation/CCN mode leads to a
relatively flat wind dependence in summer and moderately strong positive wind dependence in winter. The change-over in
wind dependence of these two modes occurs in a very small size range at the mode intersection, indicative of a different
balance of production and removal in the coarse and accumulation/CCN modes.
**4 CCN seasonal variation and covariances**
The second aspect of MBL particle behaviour considered here is a re-examination of seasonal covariances and implications
for factors influencing particle concentrations on a seasonal time scale. Use of extended time series of a seasonally-varying
species at one site allows accumulation of data over many cycles and filtering of non-seasonal factors, suppressing both shorter
term weather noise and longer term inter-annual variation. Figure 6, for example, shows the seasonal variation in N3 and CCN
(0.5%) at Cape Grim together with surface level solar UV irradiance derived from reported total ozone for Melbourne, for the
period 1978-2006. Data plotted are monthly medians and CCN (0.5%) concentrations have been offset in amplitude in the
plot, to provide approximate alignment with the N3 cycle over the autumn-winter period.
Fig. 6 illustrates the different behaviour of the three variables over summer (Dec-Feb). Whereas the UV variation is
approximately sinusoidal, N3 tends to flatten or limit and CCN concentration has a clear, cusp-like summer peak. Over the
winter part of the year (Jun-Aug) CCN concentrations are relatively constant. This presentation of annual cycles for the three
parameters, with approximate synchronisation in late autumn-winter also highlights marked differences in phase between UV,
N3 and CCN, with CCN concentrations lagging N3 by around 1-2 months leading into the summer maximum. These seasonal
cycles in general are relatively robust although for CCN in a few summers including 1997, 2004 and 2005, there was no
significant January peak.
Expressed as bivariate relationships with UV, N3 and CCN concentrations both show some seasonal hysteresis (different
response going from summer to winter to that for winter to summer) although for these aggregated seasonal data the variance
in N3 explained just by seasonal UV changes is high, around 97% (for a quadratic fit), and for CCN, UV change explains
around 87% of the seasonal monthly variance (Fig. 7). This makes no assumptions about the processes between UV irradiation
and particle number concentration, although clearly the hysteresis can be expected to be related to these processes. The
potential role of marine ecology in providing a source of reduced sulfur species that can transfer into the MBL atmosphere and
then be oxidised and condensed to particles has been very widely reported, along with possible feedback mechanisms, for
example summarised by Vogt and Liss (2009). Although a minor species by mass, aerosol-bound MSA, an oxidation product
of DMS should represent a potentially useful near-specific proxy for this biogenic source and MBL mass production.
The seasonal cycles of monthly median CCN and MSA, as shown in Fig. 8 share many significant features, particularly
relatively constant autumn-winter concentrations and the sharp summer maximum.





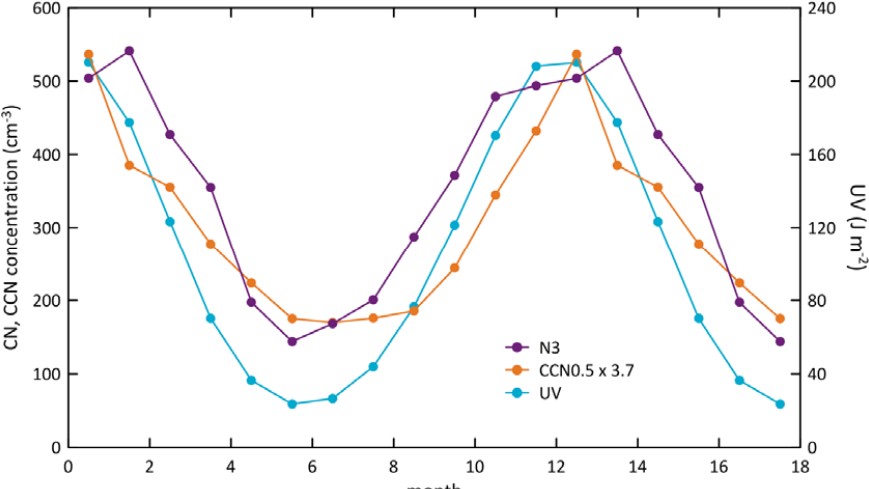

**Fig. 6. Seasonal variation in N3(1978-2005)and CCN0.5(1981-2006) with radon < 150 mBeq m$^{-3}$, and UV(1978- 2005). N3 & CCN**
**are monthly medians with 12 months data plotted as 18 months, summer is months 1,2,12,(13,14). CCN concentrations are scaled (x**
**3.7) to give a similar plotted amplitude range to N3.**

Using MSA as independent variable, the non-linear relationship for N3 shown in Fig. 9 explains a similar variance in the
seasonal cycle with UV as independent variable (r2=0.97); the very small amount of change of N3 as MSA increases during
the summer half-year (Oct-Mar) illustrates a near independence of N3 (Aitken mode) and MSA concentrations in this season).
The level of variance explained in the CCN seasonal cycle using MSA as independent variable rises to 96%, a worthwhile
improvement from the 87% explained by UV alone (Fig. 7).  MSA effectively collapses the hysteresis that was evident using
the UV/photochemistry proxy and this level of correlation with the CCN annual cycle is consistent with the hypothesised role
of marine biogenic sources in driving the seasonal pulse in MBL CCN number concentration.
As illustrated by Fig. 8, the very characteristic flat winter and sharply peaked summer underlying seasonal cycles for MSA
and CCN do not directly track the UV annual cycle, although both are very well described by the product of current month
DMS and UV irradiance. Correlation of the monthly MSA concentrations with DMS x UV gives r$^2$=0.97, and correlation of
CCN with DMS x UV r$^2$=0.96.  UV here evidently represents a simple but reasonable proxy for photochemical oxidation of
DMS.  Strong correlation was reported between ocean upper mixed layer DMS concentration and UV dose by Toole and Siegel
(2004), and more pertinently across the global ocean surface by Vallina  and Simó (2007).  At Cape Grim the seasonal pattern
of atmospheric DMS does not directly track that of UV and regression between current month values for DMS with UV
explains only around 85% of the seasonal variance.  As illustrated by Fig, 10 for MSA, the DMS cycle reaction products are
much better described by the product of current month and previous month UV (r$^2$=0.95) indicative of two dominant radiation-





limited processes with a time difference somewhere around one month. Vallina et al. (2006) consider a possible lag of between
a few days to two weeks between chlorophyll levels and an effect on CCN. Their analysis produced slightly lower correlations
between their CCN proxy and the product of chlorophyll and OH with a two week lag of their atmospheric parameters, although
their lag is referenced to chlorophyll levels and not irradiance.
The close seasonal relationship evident between MSA and the product of current and previous month UV irradiance, as shown
for MSA monthly median data in Fig. 10, and the very similar relationship for CCN and irradiance afford a simple
approximation to the dominant underlying seasonal driver, albeit stripped of all other potential modulating factors, including
biotic and physical processes.
If current month UV is used as a proxy for all photochemical production of CCN0.5, regression of CCN on UV produces a
non-photochemical offset of around 31 cm$^{-3}$ (30.7 ± 7.0, ± 1 standard error), Fig.7. The implied photochemically-derived
CCN component (from this regression) represents 74% of the observed Cape Grim summer median CCN concentration
whereas for winter the non-photochemical fraction (regression offset) dominates, contributing 65% of the observed median
CCN.
For the case where MSA is taken as a proxy for the more specific marine-biogenic photochemically-produced MBL aerosol
mass, regression with CCN0.5 (as in Fig. 9) gives a non-seasonal offset of around 48 cm$^{-3}$ (48.7 ± 2.7). In this case the
seasonal secondary component, identified with marine biogenic sources represents 60% of the observed summer median CCN
concentration (for 1981-2006 monthly median data). In winter this marine biogenic component makes no significant
contribution to the median CCN concentration.
This compares reasonably well with fractions derived for the broader Southern Ocean by Vallina et al. (2006) using correlation
based on remotely sensed chlorophyll as an indicator of marine biogenic reactive precursor sources, modelled OH and an
aerosol optical depth (AOD) fraction proxy for CCN. These authors estimate a marine biogenic fraction of 80% of the proxy
CCN in summer falling to 35% in winter, which compares better with the fractions derived above for the in-situ CCN data
using the UV/photochemical proxy, than those using MSA, a more direct proxy of marine biogenic secondary aerosol. There
are several possible explanations for differences in the two approaches including the comparison of surface observations with
column integrals. Also the remotely sensed proxy CCN involves conversion of AOD to fine fraction aerosol volume to CCN
number concentration; the accuracy of the physical representation of the paramaterisations and of the overall proxy in
representing the in-situ CCN population across the broader ocean environment have had extremely limited validation (e.g see
Gasso and Hegg 2003, Hegg and Kaufman 1998).






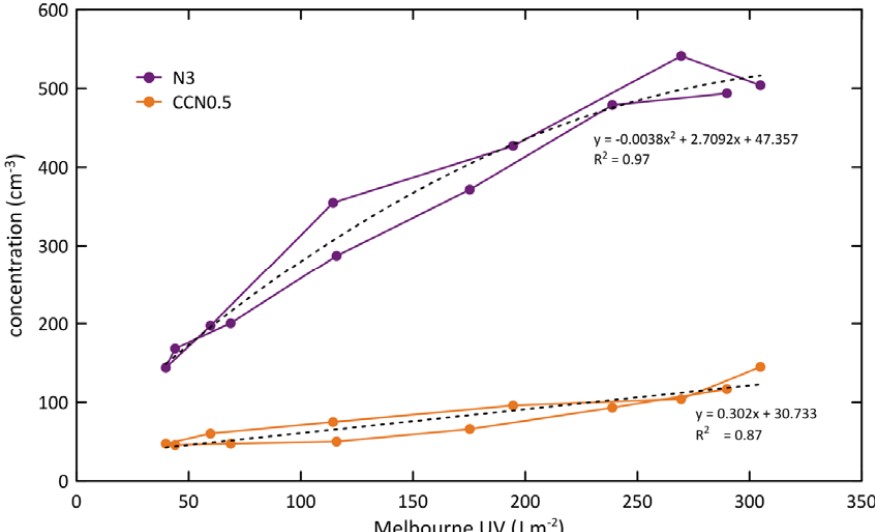

**Fig. 7. Melbourne UV 1976-2008, median CCN with radon< 150 mBeq m$^{-3}$ 1981-2006, and median CN**

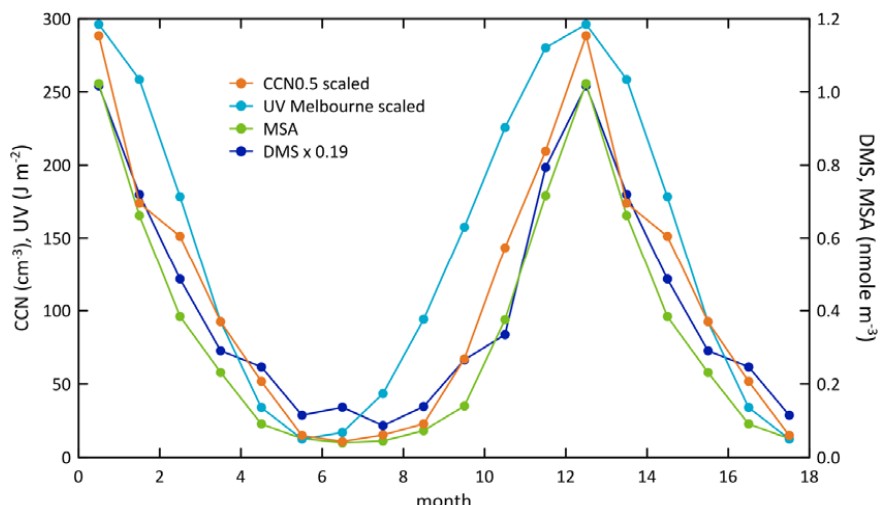

**Fig. 8. CCN and MSA (ccn median -1981-2006, [scaled], MSA median 1985-2007, UV Melbourne 1976-2008)**





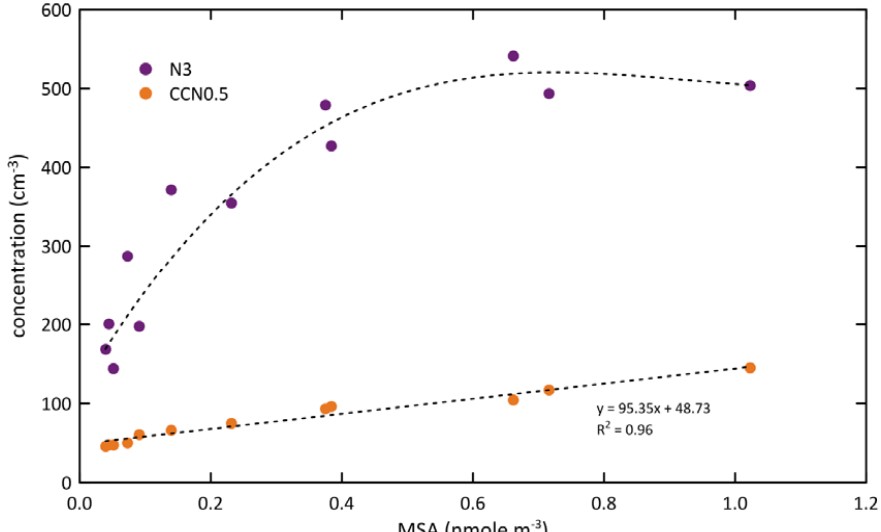

**Fig. 9. Bivariate N3 (1978-2005), CCN 1981-2006 with radon < 150 mBeq m$^{-3}$ MSA 1985-2007 (monthly medians)**

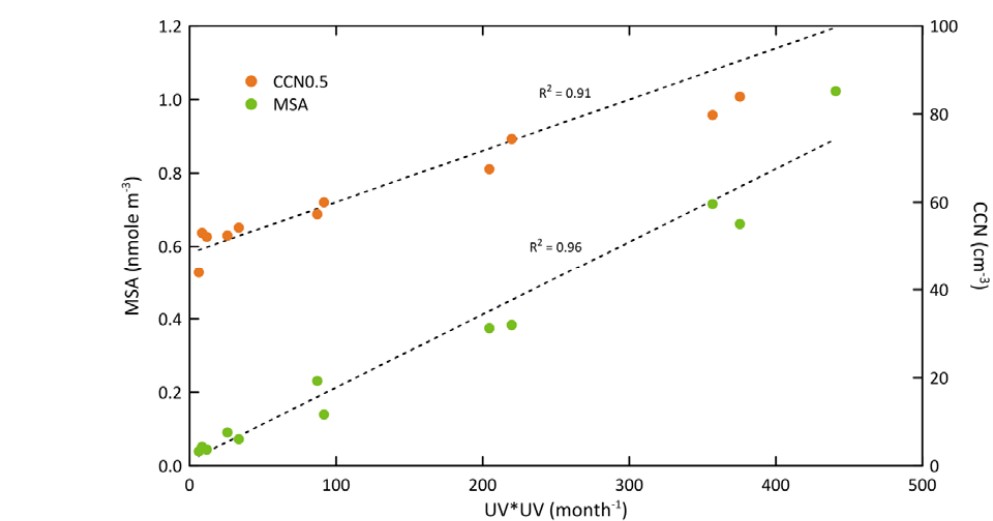

**Fig. 10. MSA and CCN0.5 dependence on UV * UV-1 month**





The offset in CCN0.5 of approximately 31 cm$^{-3}$ at zero UV levels (Fig. 7) represents a non-photochemical contribution that
should include any water-active primary particles including wind generated sea-salt and any primary organic/biogenic material
not seasonally linked to the summer pulse.  The CCN concentration - wind speed relationship, for CCN at 0.6% determined
over the far Southern Ocean (50-54 ºS), in mid-winter by Bigg et al. (1995) and the winter coarse mode concentration at Cape
Grim determined from size spectrometer measurements at Cape Grim as a function of wind speed, discussed above, suggest a
typical coarse wind-generated concentration around 20-27 cm$^{-3}$.  Whilst the major component in this fraction is likely to be
sea salt, it may also contain some primary biogenic aerosol material (PBAM) of marine origin, such as exopolymer gels,
organisms, insoluble organics and fragments; together these comprise a potentially important, but grossly understudied aerosol
component (e.g Jaenicke 2005, Leck and Bigg 2008).  In one measurement campaign at Cape Grim in summer 2006 Bigg
(2007) found that 9% of particles with D > 200 nm, and possibly up to 30% of particles with 80-200 nm contained PBAM
inclusions thus were potentially of importance for CCN activity. However, neither the actual impact of PBAM on CCN activity
nor the contribution at smaller sizes is certain.  Also since the only measurements are from in summer, it is impossible to gauge
the relationship between this class of particle and UV radiation; they might comprise a small subgroup of the 60% of CCN
already attributed to secondary marine biogenic sources, or alternatively be considered as part of the non-photochemical
primary particle group.
Regression of N3 and current month UV gives a non-UV (non-photochemical) offset of 47 cm$^{-3}$, assuming no correlation
between primary sources and UV (Fig. 7).  This includes the seasonally-invariant population of particles that are also active
as CCN at 0.5% (around 31 cm$^{-3}$), and a smaller population of particles that could be water active but are smaller than the CCN
activation diameter.
**5 CCN mode/source contributions and composition**
Some broad estimate of the various sources contributing to the CCN population is possible using regression procedures with
the previously indicated proxies; in this section both CCN active at 0.5% and the population of CCN active at 0.23% are
considered.  Whilst these populations substantially overlap giving a measure of redundancy, the CCN0.23 population excludes
a fraction of the smaller particles present in CCN0.5 and thus presents a slightly simpler particle microphysics picture.
For CCN0.23 the MSA proxy plus a seasonally-invariant component explains 96% of the variance in the CCN0.23 seasonal
cycle.  Wind speed and hence coarse mode salt has a very minor seasonal cycle (amplitude range around ± 2%) with a weak
peak around late winter-early spring, not in phase with MSA) and is treated as part of the invariant component; this can be
removed either before or after regression with similar results.   The contributions to CCN0.23 concentration from back-
substitution of the regression coefficients are shown in Fig. 11, for an analysis utilising MSA, a coarse mode based on the
observed wind speed relationship and in this case one "other", so far unspecified component.
While correlation between the underlying CCN and MSA annual cycles has been known for some time (e.g. Ayers and Gras
1991) and has giving significant insights into the MBL's annual sulfur cycle, it is significant that the MSA-like component,





taken here as a proxy for marine biogenic sources, only dominates the Southern Ocean CCN population for the 3 summer
months; the CCN0.23 component with the largest average contribution overall (from this analysis) is the "other" component
(representing 46% of CCN0.23 annually, or ~28 cm$^{-3}$); the wind-generated coarse mode varies in importance from 2$^{nd}$ to 3$^{rd}$
most important depending on the time of year.   This wind-generated coarse mode is 2$^{nd}$ most important from March-October
(autumn through to mid-spring), when the seasonal pulse in biogenic activity leads to dominance by the MSA proxy, or marine
biogenic source.
For CCN0.5 multiple linear regression with two major proxies, MSA as described above with CCN0.23 for MBL sources and
an additional proxy for Aitken mode particles (N3) explains 97% of the annual cycle monthly variance.   This produces a
seasonally-invariant component of around 38 cm$^{-3}$ and, as for CCN0.23, this is further partitioned into two components
"coarse" and "other" in Fig. 12.  In this case the MSA proxy represents the largest single CCN0.5 component for 5 months of
the year but it still only provides greater than 50% of the CCN for three months (Fig. 12).   From this analysis annual average
MSA, coarse (sea salt) and "other" proxies explain approximately equal annually-averaged contributions to CCN0.5 (26%,
28.5% and 27%).
Some contribution to CCN0.5 and at greater supersaturations, is expected from larger particles in the Aitken mode, as shown
by Covert et al. (1998) and here also from the observed size distributions.  The present analysis gives average summer and
winter CCN0.5 concentrations of 22 cm$^{-3}$ and 7 cm$^{-3}$ due to the Aitken mode compared with the corresponding mobility-
derived size distribution estimates of 38 and 5 cm$^{-3}$.
The overall broad estimates for the number concentrations of CCN0.23 and CCN0.5 based on the present analyses and wind-
dependence of coarse mode particles is summarised in Table 1.  While the MSA-like fraction is interpreted here as arising
through secondary production of mass in the MBL particularly nssSO4 and MSA it could include organic material emitted and
photo-reacted with a seasonal cycle similar to that of MSA production.  Likewise, for the Aitken mode contribution based on
the N3 proxy, composition is more likely to reflect nucleation and growth in the free troposphere (Bigg et al. 1984, Raes 1995)
and will potentially include some long-range transported precursors.
A lack of convincing mass-composition "closure" for the CCN mode (100-300 nm) in the Southern Ocean MBL aerosol
remains an issue, with data on size-dependent composition for this mode relatively sparse, particularly data on either soluble
or insoluble organics; there is also a strong bias towards composition data relating  to the spring-summer period and little for
winter.





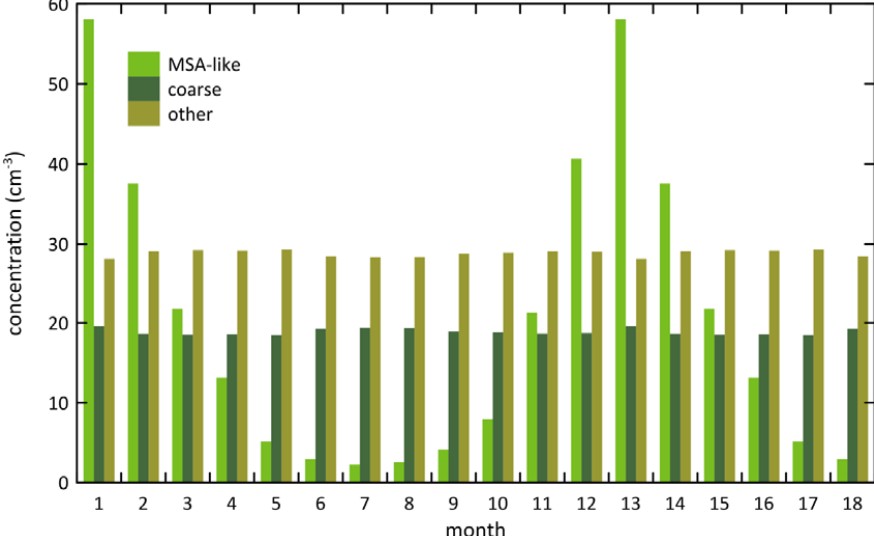

**Fig. 11. CCN.23 population based on MSA-CCN.23 regression with seasonally-invariant factor separated into a wind-generated**
**"coarse" contribution and one "other" invariant factor.**

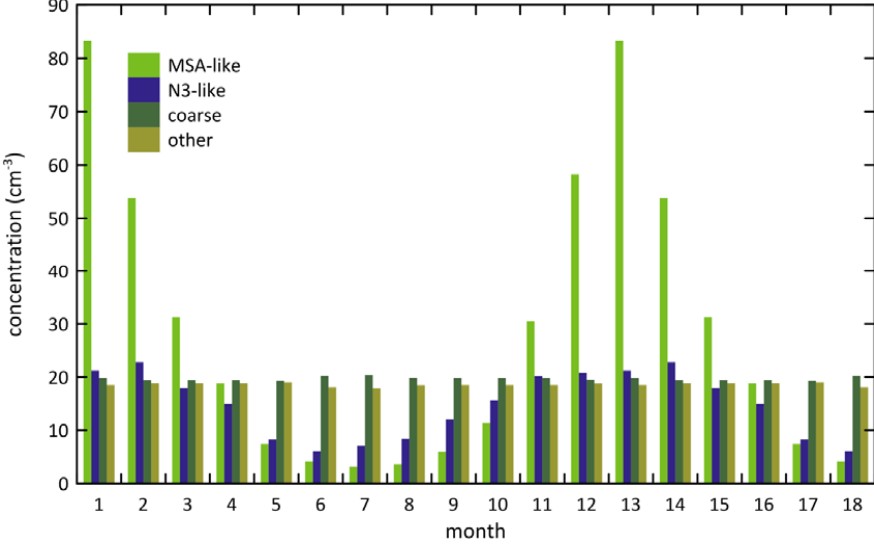


**Fig. 12. Multiple linear regression for CCN at 0.5% supersaturation as a function of 4 parameters**



**Table 1 Fractional contribution to CCN at 0.23% and 0.5% supersaturation from current analyses.**

|         |        | Aitken-like | MSA-like | coarse (SS) | other (LRT) |
|---------|--------|-------------|----------|-------------|-------------|
| CCN0.23 | summer |             | 48       | 21          | 31          |
|         | winter |             | 5        | 38.5        | 56.5        |
|         | annual |             | 23       | 31          | 46          |
| CCN0.5  | summer | 17.5        | 51.5     | 16          | 15          |
|         | winter | 14.5        | 7.5      | 41          | 37          |
|         | annual | 18.5        | 26       | 28.5        | 27          |


Single particle analyses for this size range by Gras and Ayers (1983) showed dominance of sulfate-containing particles during
summer at Cape Grim (external mixture with 95% sulfate-containing, internally 11% non-sulfate by volume, ~1%
undetermined and the residual sea-salt). This method utilised a barium thin film test for sulfate and the barium salt of MSA is
soluble, so should not appear in the sulfate fraction. The method was not specifically tested for MSA interference so cannot
be ruled out, but this suggests the non-sulfate fraction was most likely MSA. In summer at Cape Grim impactor collections
by Cainey (1997) for this size range indicate MSA close to 20% of sulfate mass loading.
During ACE-1 in spring-summer (Nov-Dec) Middlebrook et al. (1998) identified an approximately 10% organic fraction in a
size range that includes the CCN mode (D > 160 nm) but only in conjunction with sea-salt particles, and no pure organic
particles. MOUDI samples from Hueber t et al. (1998) show major non-sea-salt soluble species of sulfate, ammonium and
MSA. For the CCN-size range the missing mass fraction was 10-47% of the measured mass but uncertainty was an equivalent
magnitude. Indirect determinations for accumulation mode particles from ACE-1 (Nov-Dec) using an H-TDMA, Berg et al.
(1998), Covert et al. (1998), are consistent with sulfate/bisulfate and a small, < 10%, insoluble content. A few isolated
measurements in February 2006 by Fletcher et al. (2007), using a VH-TDMA system are also consistent with sulfate/bisulfate
as the major hygroscopic fraction but suggested variability in composition and an insoluble content ranging from 0-60%.
Cainey (1997) also determined soluble ion composition mass size distributions for the size range 100-300 nm from 1993-1994
using a MOUDI sampler and showed the predominance of sulfate and ammonium in the non-sea salt fraction over all seasons
and MSA reaching about 20% of sulfate mass loading in summer (a similar pattern to bulk sampling) and around 2% in winter.
Increased awareness of the magnitude, variability and potential role of an organic fraction in the MBL, as both water soluble
and insoluble fractions (e.g. O'Dowd et al. 2004, O'Dowd and Leeuw 2007) questions the predominantly inorganic-based
view. Northern Hemisphere observations have shown significant organic fractions associated with biologically active episodes
and this further highlights a lack of corresponding in-situ observations near highly biologically active regions in the southern
oceans. As discussed previously this applies also to marine primary biological aerosol, the actual role of which for CCN
concentration remains.


### 5.1 Seasonally-invariant CCN fraction

The component of the seasonally-invariant fraction named "other" in the CCN regression analyses clearly has no directly
identifiable information on the chemical composition of the CCN mode. Impactor analyses by Cainey (1997) have soluble-
ion-mass modes identifiable with both the accumulation/CCN mode and Aitken modes which comprise at least non-sea-salt
sulfate, ammonium, MSA and oxalate. For winter these size dependent data for $100 < D < 300$nm, which captures the mass
mode dominating the CCN population yield a total sulfate concentration of 24 ng m$^{-3}$ and non-sea-salt sulfate 16 ng m$^{-3}$;
including other soluble ions (ammonium, MSA, oxalate) gives a known mode non-sea-salt ion mass of 25.4 ng m$^{-3}$.
Multiple regression of the non-sea-salt sulfate annual cycle from multi-decadal high-volume sampler bulk samples at Cape
Grim, also using MSA and N3 as proxies similarly explains around 97% of the seasonal variance in non-sea-salt sulfate and
gives a seasonally-invariant component of 22 ng m$^{-3}$ (see Fig 13). Normalising the non-sea-salt ion mass from the MOUDI
distributions based on the non-sea-salt sulfate mode mass (16 ng m$^{-3}$) gives an expected mean (winter) accumulation/CCN
mode mass of 35 ng m$^{-3}$ for the measured non-sea-salt ion fraction. Based on the residual MSA concentration in this winter
CCN mode, only a small fraction (~2%) of this summed accumulation/CCN mode mass would be attributable to the MSA
source.
The multiple linear regressions for CCN0.23 and CCN0.5 provide two estimates of the non-seasonal component in the
accumulation/CCN mode number concentration. After removing the measured wind-speed based coarse mode concentration,
which itself has only very weak seasonal dependence and averages around 20 cm$^{-3}$ for CCN0.5 and 19 cm$^{-3}$ for CCN0.23, the
winter mean residual or "other" fraction is 28 cm$^{-3}$ based on CCN0.23 and 18 cm$^{-3}$ based on CCN0.5. Of these two estimates
CCN0.5 is considered the more accurate, given an increasing uncertainty in measurement with decreasing supersaturation in
the static CCN counter, and lower sample frequency for CCN0.23 in the Cape Grim CCN program.
These CCN number concentrations are readily converted to an equivalent mass concentration for any given particle density,
using size distributions determined at Cape Grim by mobility analysis. The average summer size distribution with a density
of 1.77 g cm$^{-3}$ (corresponding to ammonium sulfate/bisulfate) gives corresponding mass loadings of 37 ng m$^{-3}$ for CCN0.5 and
62 ng m$^{-3}$ for CCN0.23, which compare well with the median winter size distribution yields mode mass loadings of 34 ng m$^{-3}$
for CCN0.5 based and 65 ng m$^{-3}$ for CCN0.23 based. For this approximate mass closure the combined data suggest that the
majority, and possibly all of the otherwise unexplained or non-sea salt component in the seasonally-invariant factor in the CCN
concentration multiple linear regression analyses is comprised primarily of ammonium sulfate/bisulfate and minor light
organics; this implies disconnection from both the seasonal pulse in marine biogenic activity and UV annual cycle, instead
being likely associated with long range transport (LRT), potentially including an anthropogenic contribution.
Estimates by Korhonen et al. (2008) of the contribution of continental sources to CCN0.23 for 30-45 ˚S, using the chemical
transport model GLOMAP, range from ~20 cm$^{-3}$ in winter to 25 cm$^{-3}$ in summer; these values are consistent with interpreting
the constant minus coarse component estimates from multiple regression as LRT. The regressions then imply ~28 cm$^{-3}$ and
29 cm$^{-3}$ (Winter, Summer) for CCN0.23 and ~18 cm$^{-3}$ and 19 cm$^{-3}$ (Winter ,Summer) for CCN0.5 as possible LRT. Clearly



the regression analysis cannot distinguish continental precursor contributions that add to MBL sources if these react in phase
with MBL sources. In the free troposphere for example these are likely to contribute to the Aitken mode number, and mass
generation, but otherwise lose their identity. The use of a detailed model with specific sources turned off, illustrates well the
potential non-linear contribution of different sources, whereas multiple linear regression by definition identifies potential
factors contributing to CCN number variance.  A similar issue applies for example to fine primary sea spray which may
contribute to some of the kernels of CCN but not necessarily control the evolution of particles into CCN.
Recent determination of sulfur budgets for the dry, subsiding tropical marine boundary layer by Simpson et al. (2014) point to
analogous contributions to non-sea-salt sulfate from DMS production and free tropospheric entrainment (including LRT).  In
their study DMS appears responsible for only around one third of observed non-sea-salt sulfate, the remainder being attributed
to entrainment from the free troposphere.

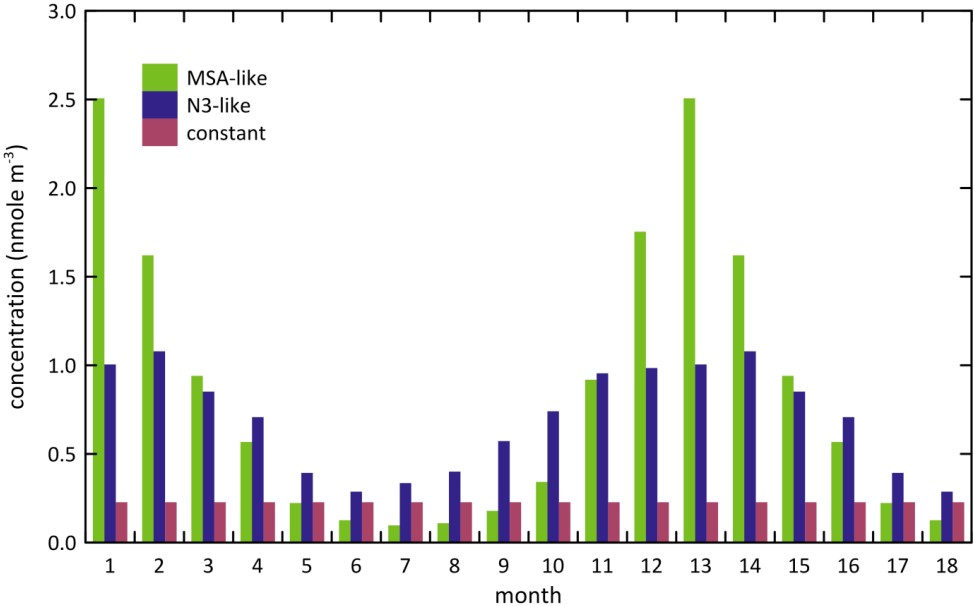

**Fig. 13. Multiple linear regression for non-sea-salt sulfate as a function of 3 parameters (MSA, N3, constant)**
**6 Conclusions**
The purpose of the present work has been to examine data on CCN concentration together with other aspects of the long term
MBL aerosol measurements at Cape Grim that should provide a useful challenge to developing regional or global numerical
models.  These analyses show that while the marine biological source of reduced sulfur appears to dominate CCN concentration
over summer, the previous strong focus on understanding sulfur cycling in the Southern Ocean MBL has somewhat





overshadowed the importance of other CCN components taken on a full annual basis. The observations show that wind-
generated coarse mode sea-salt is an important CCN component year round and from autumn through to mid spring for example
is second most important, contributing around 36% (for CCN0.23). For greater supersaturations, as expected in more
convective cyclonic systems and their associated fronts, Aitken mode particles become increasingly important as CCN. One
characteristic feature of this component is a different seasonal cycle to CCN number overall, which has a sharp summer
concentration peak. Previous regression analyses of CCN concentration have consistently indicated a non-seasonal
component, part of which can clearly be attributed to wind generated sea-salt and the remainder includes features that can be
attributed to long range transported material.
Significant contribution to the CCN population by particles from the three major size modes with seasonal change in
importance of these contributions leads to a complex wind dependence. Capturing the balance due to the combination of
different mode behaviours, as shown in the observations, is clearly a challenge to be met by numerical simulations hoping to
predict changes in CCN population in a changing climate.
**Author contributions**
J L Gras designed and led the measurement program until 2011, carried out data analysis and wrote the manuscript, M
Keywood led the measurement program since 2011 and contributed to writing the manuscript.
**Data availability**
Data are available from the authors; hourly median concentrations (CCN and CN11) from 2012-2015 are available in the
World Data Centre for Aerosols.
**Acknowledgements**
Continued support for this program through CGBAPS research funds and the personal effort of numerous Cape Grim support
staff maintaining equipment over the program lifetime is gratefully acknowledged. Ozone data determined by the Australian
Bureau of Meteorology were obtained from the World Ozone and Ultraviolet Radiation Data Centre (WOUDC) operated by
Environment Canada, Toronto, Ontario, Canada under the auspices of the World Meteorological Organization:
http://www.woudc.org/data_e.html. Radon data were provided by Wlodek Zharovski and Stewart Whittlestone from ANSTO.
Paul Selleck is thanked for carrying out aerosol analyses and Nada Derek is thanked for producing the figures in the manuscript

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
