# Peer review of "Cloud condensation Nuclei over the Southern Ocean: wind dependence and seasonal cycles"

_Atmospheric Chemistry and Physics, 2016_

## Referee Comment (RC1) · Anonymous Referee #1 · 28 Nov 2016

This study examines long-term aerosol measurements at Cape Grim with a focus on CCN sources and the sensitivity of number concentrations to wind speed, marine biological emissions, and long-range transport. There are a number of useful findings reported that will be of interest to those in the atmospheric sciences community concerned with quantifying the sources and nature of CCN in the marine boundary layer, especially over the Southern Ocean region. The paper is written well. The methods used are robust. The title and conclusions are well-supported by the data. A major strength of this work is the long-term nature of the data collected. I support publication and just had a very minor comment that the authors can choose to address if they'd like. More specifically, on Line 98, it may be likely that some readers will not be familiar

with the reference to the 'roaring forties'.

---

## Referee Comment (RC2) · K. Bigg (Referee) · 29 Nov 2016

**K. Bigg (Referee)**

keith@hotkey.net.au

Comments:

This remarkable data set has been well presented and discussed and is certain to be widely quoted. There is one aspect of it that could be the topic for future research: the influence of air trajectories on CCN and CN concentrations and their seasonal changes. There are strong temperature gradients between the Antarctic convergence zone and Tasmania. Air moving over progressively warmer waters leads to instability and convective showers, causing scavenging of the aerosol that must influence their concentrations at Cape Grim. The effect will be most important for trajectories near the southern baseline limit. In addition to this there is a zone of phytoplankton maximum
productivity, particularly active in spring and summer approximately west from Cape Grim. (e.g. Phytoplankton chlorophyll distribution and primary production in the Southern Ocean, J.K.Moore and M.R. Abbott, J. Geophys. Res. 105, C112, 28709-28722, 2000). Trajectories reaching Tasmania from directions between about 260 and 280 degrees are likely to contain higher loadings of DMS and MSA than those from further south. An indication of the locations where bacteria are most prevalent can be seen in Schnell and Vali's figure 5 (J. Atmos. Sci., 33, 1554-1564, 1976) that showed high IN concentrations extending due west from Cape Grim.

If air trajectories are found to significantly alter CCN, CN or MSA concentrations it would be important information for inclusion in global aerosol models.

The presence of a seasonally invariant CCN component that the authors have discovered is very interesting. They attribute it in part to long-range transported material. Black carbon measurements at Cape Grim or Antarctica have a strong seasonal component and are probably indicative of seasonal changes in the more general transported aerosol. As a pure speculation, I suggest oceanic microgels as the invariant CCN component. Verdugo (Ann. Reviews Marine Science, 37, 375-400, 2012) showed such an immense reservoir of these microgels in the ocean that seasonal components would be almost absent.

---

## Author Comment (AC1) · 10 Feb 2017

Comments from Reviewer 1

This study examines long-term aerosol measurements at Cape Grim with a focus on CCN sources and the sensitivity of number concentrations to wind speed, marine biological emissions, and long-range transport. There are a number of useful findings reported that will be of interest to those in the atmospheric sciences community concerned with quantifying the sources and nature of CCN in the marine boundary layer, especially over the Southern Ocean region. The paper is written well. The methods used are robust. The title and conclusions are well-supported by the data. A major strength of this work is the long-term nature of the data collected. I support publication

and just had a very minor comment that the authors can choose to address if they'd like. More specifically, on Line 98, it may be likely that some readers will not be familiar with the reference to the 'roaring forties'.

Authors Response We thank Reviewer 1 for their positive and supportive comments. Regarding the comment about the roaring forties (line 98) we have reworded the text to provide an explanation of this term.

Changes to the manuscript text Line 98-100 The Southern Ocean region upwind of Cape Grim comprises part of the "roaring forties" which are strong westerly winds that generally occur between 40 and 50 degrees south. It is a term persisting from the days of sailing ships and has a well-earned and enduring reputation for strong and persistent winds.

---

## Author Comment (AC2) · 10 Feb 2017

Comments from Keith Bigg

This remarkable data set has been well presented and discussed and is certain to be widely quoted. There is one aspect of it that could be the topic for future research: the influence of air trajectories on CCN and CN concentrations and their seasonal changes. There are strong temperature gradients between the Antarctic convergence zone and Tasmania. Air moving over progressively warmer waters leads to instability and convective showers, causing scavenging of the aerosol that must influence their concentrations at Cape Grim. The effect will be most important for trajectories near the southern baseline limit. In addition to this there is a zone of phytoplankton maximum

productivity, particularly active in spring and summer approximately west from Cape Grim. (e.g. Phytoplankton chlorophyll distribution and primary production in the Southern Ocean, J.K.Moore and M.R. Abbott, J. Geophys. Res. 105, C112, 28709-28722, 2000). Trajectories reaching Tasmania from directions between about 260 and 280 degrees are likely to contain higher loadings of DMS and MSA than those from further south. An indication of the locations where bacteria are most prevalent can be seen in Schnell and Vali's figure 5 (J. Atmos. Sci., 33, 1554-1564, 1976) that showed high IN concentrations extending due west from Cape Grim. If air trajectories are found to significantly alter CCN, CN or MSA concentrations it would be important information for inclusion in global aerosol models. The presence of a seasonally invariant CCN component that the authors have discovered is very interesting. They attribute it in part to long-range transported material. Black carbon measurements at Cape Grim or Antarctica have a strong seasonal component and are probably indicative of seasonal changes in the more general transported aerosol. As a pure speculation, I suggest oceanic microgels as the invariant CCN component. Verdugo (Ann. Reviews Marine Science, 37, 375-400, 2012) showed such an immense reservoir of these microgels in the ocean that seasonal components would be almost absent.

Authors response

We thank Dr Bigg for his very positive and supportive comments and suggestions for future work and improvements to the manuscript. Dr Bigg raises two important points Dr Biggs first point relates to a topic for future research i.e. determination of the influence of air trajectory on CN & CCN concentration and their seasonal variation; some specific dynamical and precursor spatial distribution factors are suggested. We agree fully with the Dr Bigg on the value of this topic of research, agreeing that multiple factors including regional dynamics, precursor emissions and conversion and removal are all significant. Some indication of differences in ability to accurately model CCN concentration at CG, evident with different trajectory histories within the broad "Baseline" category was shown already by Covert et al. (1998); further factors should be dis-

coverable through fixed site observations such as at CG although clearly the greatest potential for unpicking the links between dynamical and microphysical factors requires comprehensive modeling, in conjunction with widely-based observations, as for example through approaches like Korhonen et al., 2008, Spracklen et al., 2005. Again there is further potential in this area as observations, guided by the models, improve. Changes to the manuscript text –none

Dr Biggs second point relates to our demonstration of the presence of a seasonally invariant component.

Dr Bigg notes that Black carbon (BC) at Cape Grim and in the Antarctic has a strong seasonal cycle which is possibly reflective of broad scale transport of aerosol; the potential importance of microgels is advanced as a speculative source of the seasonally invariant component. We agree with the Dr Biggs observation of the seasonal cycle in BC, although note that any one specific aerosol component reflects not only dynamical factors but also specific seasonality's related to its generation and removal, where generally these processes are also size-dependent. A multifactorial analysis of the time-variability of aerosol and related surrogates for Cape Grim is in fact in progress. The potential contribution of microgels and exopolymeric gel particles to CCN activity over the Southern Ocean is indeed a fascinating issue that clearly warrants further study. Without direct extended observation records of marine gel CCN contribution (or chemical surrogate) sorting out wind or seasonal dependence in the CCN record is problematic, clearly this is a case of more observations required.

Change to the manuscript text: we broaden the range of maritime gels referenced. Line 341-344 changed to Whilst the major component in this fraction is likely to be sea salt, it may also contain some primary biogenic aerosol material (PBAM) of marine origin, such as marine gels, organisms, insoluble organics and fragments; together these comprise a potentially important, but grossly understudied aerosol component (e.g Jaenicke 2005, Leck and Bigg 2008).

References Korhonen, H., Carslaw, K. S., Spracklen, D. V., Mann, G. W., and Woodhouse, M. T.: Influence of oceanic dimethyl sulfide emissions on cloud condensation nuclei concentrations and seasonality over the remote Southern Hemisphere oceans: A global model study, Journal of Geophysical Research-Atmospheres, 113, 10.1029/2007jd009718, 2008. Spracklen, D. V., Pringle, K. J., Carslaw, K. S., Chipperfield, M. P., and Mann, G. W.: A global off-line model of size-resolved aerosol microphysics: I. Model development and prediction of aerosol properties, Atmospheric Chemistry and Physics, 5, 2227-2252, 2005.